# Comprehensive Insight into Chemical Stability of Important Antidiabetic Drug Vildagliptin Using Chromatography (LC-UV and UHPLC-DAD-MS) and Spectroscopy (Mid-IR and NIR with PCA)

**DOI:** 10.3390/molecules26185632

**Published:** 2021-09-16

**Authors:** Anna Gumieniczek, Anna Berecka-Rycerz, Emilia Fornal, Barbara Żyżyńska-Granica, Sebastian Granica

**Affiliations:** 1Department of Medicinal Chemistry, Faculty of Pharmacy, Medical University of Lublin, Jaczewskiego 4, 20-090 Lublin, Poland; anna.berecka@umlub.pl; 2Department of Pathophysiology, Faculty of Medicine, Medical University of Lublin, Jaczewskiego 8b, 20-090 Lublin, Poland; emilia.fornal@umlub.pl; 3Department of Biochemistry, Faculty of Medicine, Medical University of Warsaw, Banacha 1, 02-097 Warsaw, Poland; bzyzynska@wum.edu.pl; 4Microbiota Lab, Department of Pharmacognosy and Molecular Basis of Phytotherapy, Faculty of Pharmacy, Medical University of Warsaw, Banacha 1, 02-097 Warsaw, Poland; sgranica@wum.edu.pl

**Keywords:** vildagliptin and stability, pH and oxidative conditions, high temperature and humidity, kinetics of degradation, interactions with excipients, LC-UV, UHPLC-DAD-MS, mid-IR and NIR with PCA

## Abstract

During forced degradation, the intrinsic stability of active pharmaceutical ingredients (APIs) could be determined and possible impurities that would occur during the shelf life of the drug substance or the drug product could be estimated. Vildagliptin belongs to relatively new oral antidiabetic drugs named gliptins, inhibiting dipeptidyl peptidase 4 (DPP-4) and prolonging the activities of the endogenous incretin hormones. At the same time, some gliptins were shown as prone to degradation under specific pH and temperature conditions, as well as in the presence of some reactive excipients. Thus, forced degradation of vildagliptin was performed at high temperature in extreme pH and oxidative conditions. Then, selective LC-UV was used for quantitative determination of non-degraded vildagliptin in the presence of its degradation products and for degradation kinetics. Finally, identification of degradation products of vildagliptin was performed using an UHPLC-DAD-MS with positive ESI. Stability of vildagliptin was also examined in the presence of pharmaceutical excipients, using mid-IR and NIR with principal component analysis (PCA). At 70 °C almost complete disintegration of vildagliptin occurred in acidic, basic, and oxidative media. What is more, high degradation of vildagliptin following the pseudo first-order kinetics was observed at room temperature with calculated k values 4.76 × 10^−4^ s^−1^, 3.11 × 10^−4^ s^−1^, and 1.73 × 10^−4^ s^−1^ for oxidative, basic and acidic conditions, respectively. Next, new degradation products of vildagliptin were detected using UHPLC-DAD-MS and their molecular structures were proposed. Three degradants were formed under basic and acidic conditions, and were identified as [(3-hydroxytricyclo- [3.3.1.13,7]decan-1-yl)amino]acetic acid, 1-{[(3-hydroxytricyclo[3.3.1.13,7]decan-1-yl)amino]acetyl}-pyrrolidine-2-carboxylic acid and its *O*-methyl ester. The fourth degradant was formed in basic, acidic, and oxidative conditions, and was identified as 1-{[(3-hydroxytricyclo[3.3.1.13,7]-decan-1-yl)amino]acetyl}pyrrolidine-2-carboxamide. When stability of vildagliptin was examined in the presence of four excipients under high temperature and humidity, a visible impact of lactose, mannitol, magnesium stearate, and polyvinylpirrolidone was observed, affecting-NH- and CO groups of the drug. The obtained results (kinetic parameters, interactions with excipients) may serve pharmaceutical industry to prevent chemical changes in final pharmaceutical products containing vildagliptin. Other results (e.g., identification of new degradation products) may serve as a starting point for qualifying new degradants of vildagliptin as it is related to substances in pharmacopoeias.

## 1. Introduction

Gliptins constitute a class of drugs increasingly used for the treatment of type 2 diabetes mellitus, inhibiting dipeptidyl peptidase 4 (DPP4), the enzyme that inactivates the incretin hormones such as glucagon-like peptide 1 (GLP1) and glucose dependent insulinotropic polypeptide (GIP). GLP1 and GIP serve as important prandial stimulators of insulin secretion and regulators of blood glucose concentration. Thus, inhibition of DPP4 by gliptins prolongs the activities of endogenous GLP1 and GIP, decreasing the elevated blood glucose in diabetic patients [1]. The DPP-4 inhibitors can be divided as peptidomimetics which mimic the DPP-4 enzyme and no peptidomimetic agents. Vildagliptin is a substrate-mimicking inhibitor containing cyanopyrrolidine motif [2]. It is rapidly absorbed and quickly cleared from plasma, and required to be administered twice daily as compared to once daily dosing for some other gliptins [3]. The chemical structure of vildagliptin is depicted in Figure 1.

A few HPLC and LC/MS methods were described in the literature for determination of vildagliptin in one-component formulations [4,5] or plasma [6,7]. Besides, HPLC and LC/MS methods were developed for determination of vildagliptin in the presence of other drugs in two- or three-component mixtures [8,9,10,11] and biological fluids [11,12,13,14]. Only a few authors performed forced degradation of vildagliptin as a part of validation of HPLC or LC/MS procedures to confirm stability-indicating properties of these methods [15,16,17]. What is more, in the literature there is no any report on kinetic aspects of stability of vildagliptin which are very important in the pharmaceutical industry [18]. Thus, the first goal of the present study was to determine the drug stability over time, taking into account the impact of extreme pH and oxidative conditions using a sufficiently selective LC-UV method. Additionally, identification of degradants and understanding the degradation pathways of active pharmaceutical ingredients (APIs) play a crucial role in the rational drug design, and are extremely important for their safety and potency [19]. Thus far, just a few papers on degradants or impurities of vildagliptin were published [15,20,21,22]. In the study of Barden [15], the basic and oxidative conditions were applied and one degradation product was characterized using LC-MS. Recently, Arar et al. [20] reported six degradation products of vildagliptin in acidic, basic and oxidative conditions. Besides, the groups of Kumar [21] and Al-Sabti and Harbali [22] reported some impurities due to the production of vildagliptin itself. Bearing in mind a small number of published papers in this area and the necessity of detecting all possible degradants of vildagliptin, the second goal of the present study was to performed accelerated degradation of vildagliptin and deeply analyze the stressed samples using UHPLC-DAD-MS method. In addition, many guidelines emphasize the importance of testing stability of APIs in final pharmaceutical formulations. Now, it is well known that active substances can react with excipients, mainly due to hydrolysis and redox reactions [23,24]. In the literature, some papers on visible interactions of other gliptins, e.g., sitagliptin, with pharmaceutical excipients were published [25]. Meanwhile, chemical stability of vildagliptin in the presence of excipients was not studied extensively so far. Only one paper in this area was found in the literature [26], where compatibility of vildagliptin with silicon dioxide, carbopol, microcrystalline cellulose, polyvinylpyrrolidone, and magnesium stearate was studied. Thus, the next goal of the present study was to examine stability of vildagliptin in the presence of four excipients of different reactivity, i.e., lactose (LAC), mannitol (MAN), magnesium stearate (MGS) and polyvinylpirrolidone (PVP). The solid mixtures were stressed with high temperature and high humidity, and analyzed using mid-IR and NIR methods together with chemometric assessment using Principal Component Analysis (PCA).

## 2. Results and Discussion

### 2.1. Optimization and Validation of LC-UV Method for Quantitative Measurements of Vildagliptin

Optimization and validation of quantitative LC-UV method was performed, involving robustness, selectivity, linearity, precision, and accuracy. The most important parameters of validation are shown in Table 1. It was found that simple mobile phase containing 2 mM ammonium acetate and acetonitrile at 80:20 (*v/v*) ratio was sufficiently effective for separation of the peaks of interest in a reasonable time, as well as, for reduction of the peak tailing. The chromatograms showed that the peaks of vildagliptin were free from interferences of these from degradation products. All detailed results from these experiments were presented in Appendix A.

### 2.2. Degradation of Vildagliptin in Solutions and Degradation Kinetics

According to the literature, the highest degradation of vildagliptin occurred in 6% H_2_O_2_ at room temperature (degradation above 25% after 30 min) [17]. It was also observed that vildagliptin was sensitive to degradation in 0.01 M NaOH at 60 °C (above 10% of degradation after 30 min), but rather stable in 1M HCl at 60 °C (degradation below 5% after 2 h). Our study showed that vildagliptin could degrade in wider pH range, as well as, in oxidative conditions, especially at high temperature. At 70 °C its degradation in 1M HCl was almost 85% after 210 min. What is more, the drug was completely degraded in 1M NaOH and 3% H_2_O_2_ after 60 min. Therefore, our experiment was performed once more, using the same solutions (1M HCl, 1M NaOH and 3% H_2_O_2_) at controlled room temperature (23 °C). Under these conditions, degradation of vildagliptin was shown as 59.28% (1M HCl) and 84.33% (1M NaOH) after 240 min. In 3% H_2_O_2_ the level of degradation gained 87.04% after 180 min, showing the lowest stability of vildagliptin in oxidative conditions (Table 2). Kinetic parameters were calculated for all degradation conditions at 23 °C and acidic conditions at 70 °C. The logarithm of concentration of no degraded vildagliptin was stronger correlated than the concentrations of no degraded vildagliptin with time of degradation, confirming the pseudo-first-order kinetics for these processes with all R^2^ values above 0.96. At 23 °C, the calculated rate constants of degradation (k) were at the level of 10^−4^ s^−1^, while the degradation time of 50% (t_0.5_) varied from 1.11 h in 1M HCl through 0.62 h in 1M NaOH to 0.40 h in 3% H_2_O_2_, confirming the quickest degradation of the drug in oxidative medium. In 1M HCl at 70 °C, the calculated k value was also at the level of 10^−4^ s^−1^, while corresponding t_0.5_ equaled 0.72 h (Table 2). The obtained results were also depicted as the xy diagrams in Figure 2. Due to the lack of other scientific data in this area, the results presented here are a valuable supplement to the literature resources.

### 2.3. Identification of Vildagliptin Degradation Products by UHPLC-DAD-MS

In the literature, four degradation products related to production of vildagliptin were described as IMPs A–D [27]. Besides, a new degradant was detected during purification of crude vildagliptin with ethyl methyl ketone, i.e., IMP E, which easily decomposed to a stable IMP F [21]. In the study of Barden et al. [15], the oxidative and basic degradation of vildagliptin was reported. When LC-MS analysis was applied for the stressed samples, the main degradation product was detected at *m*/*z* 154 (DP 1). Recently, Arar et al. [20] reported six degradants of vildagliptin, i.e., IMP B and five new compounds (DP 2–6) which were formed in acidic, basic or oxidative conditions. The related substances of vildagliptin which were previously described in the literature are presented in Table 3.

Our UHPLC-DAD-MS experiments allowed to separate and identify four degradants of vildagliptin, namely Compounds A–D. Only one of which, i.e., Compound B was described previously as IMP B [20,27]. The chemical structures and *m*/*z* values for all degradants identified in the present study are presented in Table 4.

Compound A was detected when vildagliptin was stressed in acidic and basic conditions at both, room (23 °C) and high (70 °C) temperature, and showed a molar mass 225.29 g/mol (*m*/*z* 226 as [M + H]^+^). It was supposed that the pyrrolidine-2-carbonitrile motif was left from the structure of vildagliptin to produce the corresponding carboxylate (Figure 3). As far as Compound B was concerned, it was detected in basic and oxidative conditions, similarly to the results from the literature [20] and for the first time, in acidic conditions as well. It showed a molar mass 321.42 g/mol (*m*/*z* 322 as [M + H]^+^) and was probably formed by hydrolysis of cyano group of vildagliptin into the amide one. Thus, our results pointed to hydrolysis of the cyano group of vildagliptin in basic, oxidative, as well as, acidic conditions. Further hydrolysis of the amide group of Compound B afforded the corresponding carboxylic acid, leading to Compound C which showed a molar mass 322.41 g/mol (*m*/*z* 323 as [M + H]^+^). Since methanol was present in our stressed samples, methyl ester of respective acid was also detected (Compound D).

It was interesting to observe that other degradants of vildagliptin that had been reported previously [15,20], were not observed in the present study. In addition, it seemed reasonable to discuss more deeply the structures of our Compound C and the degradant DP 4 from the literature [20]. DP 4 with pseudomolecular ion at *m*/*z* 323.6 was detected in the samples of vildagliptin stressed in basic conditions. Bearing in mind its retention time (t_R_) in respective LC chromatograms, it was relatively more polar than IMP B. In the present study, Compound C with *m*/*z* 323 was detected in the samples of vildagliptin stressed in basic, as well as, acidic conditions. On the contrary to DP 4, Compound C was shown to be relatively less polar than Compound B (IMP B), because of its longer t_R_ during our LC-MS experiments. Thus, the chemical structure of Compound C, different than that that described for DP 4 was proposed. Respective UHPLC data of the detected Compounds A–D are given in Table 5. Chromatograms (ion current, BPC MS+) of the analyzed samples and vildagliptin pure standard are presented in Figure 4. MS and UV/VIS spectra of the detected Compounds A–D are shown in Figure 5, Figure 6, Figure 7, Figure 8 and Figure 9.

There are some other minor peaks at our BPC chromatograms of the analyzed samples, namely Imps 1–3 (Table 5), probably from impurities present in our vildagliptin standard. Their MS and UV/VIS spectra are shown in Appendix A.

### 2.4. Stability of Vildagliptin in the Presence of Excipients

It is known that APIs can react with excipients in final pharmaceutical formulations [23,24]. Thus, stability of vildagliptin was also examined in the presence of four excipients of different reactivity, i.e., lactose (LAC), mannitol (MAN), magnesium stearate (MGS) and polyvinylpirrolidone (PVP), under high temperature and high humidity (60 °C and 70% RH). Finally, the mixtures were analyzed using mid-IR and NIR spectroscopy together with chemometric assessment. In details, we used Principal Component Analysis (PCA) to identify specific wavenumbers discriminating the changes of vildagliptin due to interactions with excipients and accelerated degradation. PCA is very useful when IR spectra are analyzed, because contribution of each original spectral variable to each PC becomes visible in the loadings’ spectra. As a consequence, it allows identifying important spectral bands accounting for the most significant differences between samples.

From the obtained mid-IR and NIR spectra it was seen that vildagliptin itself did not visibly change at high temperature and high humidity. Also, the spectra of individual excipients did not show changes after the stress (Table 6).

When mid-IR and NIR spectra were analyzed by PCA, visible separation of individual vildagliptin and each individual excipient was observed. In addition, the stressed mixtures were separated from the no stressed ones as well as from all individual compounds (Figure 10A–D and Figure 11A–D).

#### 2.4.1. Vildagliptin and LAC

After mixing vildagliptin and LAC, the signals of N-H and O-H vibrations of vildagliptin at 3294 cm^−1^ were overlapped with a broad peak of LAC at 3000–3400 cm^−1^. In addition, the peak of vildagliptin at 1040 cm^−1^ disappeared (Figure 12A). Additive changes were seen in the mixture after accelerated degradation, e.g., disappearance of the peak corresponding to N-H bending vibrations of vildagliptin at 1495 cm^−1^. Thus, it was supposed that -NH- group of vildagliptin could interact with LAC at high temperature and humidity. Previously, similar interactions with LAC were observed for sitagliptin [25] and for many APIs containing primary and secondary amine groups [28]. In addition, a sharp band of CO group from vildagliptin at 1658 cm^−1^ changed its shape and was extended from 1500 cm^−1^ to 1700 cm^−1^ in the stressed mixture (Figure 12B). When mid-IR spectra were analyzed by PCA, PC1 and PC2 explained 51.50 and 29.60% of variability. The loading spectrum of PC1 was dominated by the bands at 980–1050 cm^−1^ and 3000–3300 cm^−1^. These bands mainly showed spectral differences between vildagliptin and LAC themselves, but to some extent, differences due to degradation involving the amine group of the drug as well. In addition, other regions of variability were observed at 2800–2950 cm^−1^ (both PC1 and PC2 bands were negative) and 1450–1350 cm^−1^ (PC1 band was negative while PC2 band was positive). These bands visibly showed differences due to accelerated degradation of vildagliptin in the presence of LAC. What is more, the band at 1550–1630 cm^−1^ in the PC2 loading spectrum served as evidence of interactions involving the carbonyl group of vildagliptin (Figure 12C).

When NIR spectra were analyzed by PCA, PC1 explained 99.30% of variability while PC2 0.494%. PC1 variability was mainly connected with obvious spectral differences between vildagliptin and LAC but some interactions with this excipient could also be confirmed in respective spectra (Figure 13A,B). According to PC1 values, the main source of variability was connected with an increase of absorbance in the region 6000–7000 cm^−1^ (the first overtones of C-H, N-H and O-H) with a characteristic decrease at 6400 cm^−1^. In addition, an increase of absorbance in the region 4700–5200 cm^−1^ (the first and second overtones of C=O and the C-H and O-H combinations) was observed. When PC2 loadings were considered, the main source of variability was connected with the wavenumbers 5300 cm^−1^, 4700 cm^−1^ (the N-H and O-H combinations) and 6500 cm^−1^ which could reflect changes due to interactions of vildagliptin with LAC (Figure 13C).

#### 2.4.2. Vildagliptin and MAN

After mixing vildagliptin with MAN, the signal of N-H and O-H vibrations of vildagliptin at 3294 cm^−1^ were overlapped with respective peaks of MAN (3200–3300 cm^−1^). In addition, a band of MAN at 1090 cm^−1^ deteriorated to 1076 cm^−1^, probably due to hydrogen bond formation between its -OH groups and -NH- group of vildagliptin (Figure 14A). After stressing with high temperature and humidity, the peak corresponding to a sharp band of CO from vildagliptin at 1658 cm^−1^ changed its shape (Figure 14B). For the mid-IR spectra, PC1 and PC2 explained 63.30 and 21.50% of variability. The loading spectrum of PC1 was dominated by a broad band at 3000–3500 cm^−1^ showing spectral differences that occurred in the mixture of vildagliptin and MAN in these regions of the spectra. On the other hand, the bands at 1450–1350 cm^−1^ where PC1 was positive and PC2 was negative, and the broad band at 1650–2000 cm^−1^ in PC2 loading spectrum indicated some changes due to accelerated degradation of vildagliptin in the presence of MAN (Figure 14C).

When NIR spectra were analyzed, a decrease of characteristic bands at 6500 cm^−1^ and 4700 cm^−1^ was observed, as a result of interactions of vildagliptin with MAN at high temperature and humidity (Figure 15A,B). Inspecting PC values, it was concluded that PC1 explained 99.30% of variability while PC2 0.641%. According to PC1 loadings, the main source of variability was connected with an increase of absorbance in the region 5900–7000 cm^−1^ and a characteristic decrease at 6400 cm^−1^. When PC2 value was considered, the main source of variability was seen in the 5000–5800 cm^−1^ region (the first and second overtones of C=O and the first overtones of C-H) (Figure 15C).

#### 2.4.3. Vildagliptin and MGS

After mixing vildagliptin and MGS, a broad band of vildagliptin at 3294 cm^−1^ was still clearly seen. Similarly, the band due to -COO- stretching of MGS at 1577 cm^−1^ was not affected. Thus, the lack of interactions via hydrogen bonding between -NH- group of vildagliptin and -COO- group of MGS was confirmed (Figure 16A). When the mixture was stressed with high temperature and humidity, the peak due to the secondary amine of vildagliptin did not change. However, other characteristic bands changed their shapes or disappeared. It was clearly observed for bands of vildagliptin due to C=O vibrations at 1658 cm^−1^ and OH vibrations at 1404 cm^−1^. It was also observed that the bands of MGS at 1577 cm^−1^ and 1505 cm^−1^ changed their shaped or disappeared (Figure 16B). When mid-IR spectra were examined by PCA, the loading spectra of PC1 (50.80% of variability) and PC2 (25.20% of variability) were dominated by the bands at 1400–1600 cm^−1^ and 2800–3000 cm^−1^, reflecting the differences in the spectra of individual vildagliptin and MGS. The major difference between these PCs was that the intensities of the PC1 bands were correlated positively while PC2 negatively. In addition, PC2 plot showed a new region of variability from 1600 cm^−1^ to 1700 cm^−1^. It confirmed the changes due to degradation involving CO group of vildagliptin. On the other hand, compatibility of vildagliptin with MGS after storing at 25–40 °C/60–75% RH for four weeks was observed in the study of Sravani et al. [26]. These discrepancies could be, at least in part, due to milder experimental conditions and shorter degradation time in these previous experiments.

It was observed that the peaks of vildagliptin at 6500 cm^−1^ and 4700 cm^−1^ visibly disappeared in the NIR spectrum of the stressed mixture (Figure 17A,B). Inspecting PCs values for NIR spectra it was concluded that PC1 explained 99.60% of variability while PC2 0.303%. According to PC1 value, the main sources of variability were connected with increases of absorbance in the regions 4100–4300 cm^−1^ and 7100–7300 cm^−1^ (the first overtones of N-H and O-H). However, there was a difference within the regions 4000–4900 cm^−1^ (the combinations of NH and O-H) and 6000–6800 cm^−1^ (the first overtones of N-H and C-H) where the PC1 values were negative. At the same time, visible differences in these regions were seen in the spectra of the stressed mixture (Figure 17B). When PC2 values were considered, the main source of variability was connected with the region 4880–5300 cm^−1^ (the first and second overtones of C=O) (Figure 17C). At the same time, the peaks of vildagliptin at 6500 cm^−1^ and 4700 cm^−1^ visibly disappeared in the spectrum of the stressed mixture (Figure 17B). Thus, some parts of variability in the stressed mixture could be explained by both PC1 and PC2 values.

#### 2.4.4. Vildagliptin and PVP

After mixing vildagliptin and PVP, the band due to signals of OH and NH groups of the drug at 3294 cm^−1^ was still clearly seen. However, the signal of C=O group of vildagliptin at 1658 cm^−1^ was overlapped with the peak of PVP at 1650 cm^−1^ (Figure 18A). When the mixture of vildagliptin and PVP was treated with high temperature and humidity, the changes concerning the peak at 3294 cm^−1^ were observed (Figure 18B). Based on these changes we supposed that the amine group of vildagliptin could be affected in the presence of PVP. In addition, the overlapped band of C=O stretching vibration of vildagliptin at 1658 cm^−1^ was visibly broadened. Thus, the carbonyl group of vildagliptin could also be affected in the presence of PVP and accelerated degradation. The loading spectra of PC1 (50.60% of variability) and PC2 (39.50% of variability) were dominated by the bands at 2800–2950 cm^−1^ that scored positively for PC2 and negatively for PC1 values. In addition, the bands in the 3300–3600 cm^−1^ and 1200–1330 cm^−1^ regions were clearly seen (Figure 18C). All these PCs showed differences between the spectra of individual vildagliptin and individual PVP. However, PC2 plot showed news regions of variability at 3000–3300 cm^−1^ and 1600–1700 cm^−1^ (Figure 18C). Thus, some changes concerning the amine and carbonyl groups of vildagliptin due to the presence of PVP and accelerated degradation were supposed. On the other hand, compatibility of vildagliptin with PVP was confirmed previously [26]. However, much lower temperature and shorter time was then used for accelerated degradation and probably, some interactions had not been occurred previously.

When the degraded mixture was analyzed by NIR method, visible changes occurred at 6400–6500 cm^−1^ and 5000–5100 cm^−1^ (Figure 19A,B). Inspecting PCs values for vildagliptin and P it was concluded that PC1 explained 99.60% of variability while PC2 0.320%. PC1 and PC2 loadings plots were dominated by the bands at 4600–4700 cm^−1^, 4100–4400 cm^−1^ and 1600–1700 cm^−1^ where PC1 scored negatively while PC2 scored positively, and at 4700–5300 cm^−1^ where both PCs scored positively. The most of variability showed spectral differences between vildagliptin and PVP themselves. However, characteristic bands in PC loadings confirmed the changes due to chemical interactions, concerning vildagliptin (6500 cm^−1^) and PVP (5150 cm^−1^).

## 3. Materials and Methods

### 3.1. Reagents and Chemicals

Vildagliptin, lactose (LAC), mannitol (MAN), magnesium stearate (MGS) and polyvinylpirrolidone (PVP) from Sigma-Aldrich (St. Louis, MO, USA) and phenacetin (I.S. for LC-UV method) from Marcmed (Lublin, Poland) were used. Solvents for chromatography were of HPLC or LC-MS grade, and were purchased from Merck KGaA (Darmstadt, Germany) or Sigma-Aldrich. Deionized water was produced at our laboratory with a Simplicity UV Water Purification System from Merck-Millipore (Burlington, MA, USA). Other chemicals were of analytical grade and were supplied by POCh (Gliwice, Poland) and Sigma-Aldrich. Galvus^®^ tablets containing 50 mg of vildagliptin were produced by Novartis Europharm (Basel, Switzerland).

### 3.2. Stability of Vildagliptin in Solutions

#### 3.2.1. Accelerated Degradation

Equal volumes of 1 mL from the stock solutions of vildagliptin (1 mg/mL) were dispensed to small glass dishes with matching glass stoppers and mixed with 1 mL of respective stressors (1M HCl, 1M NaOH and 3% H_2_O_2_). The dishes were tightly closed and stored in an air-conditioned room at 23 °C or placed in a thermo-stated water bath set at 70 °C. The samples were analyzed subsequently after 30, 60, 90, 120, 150, 180, 210 and 240 min.

#### 3.2.2. Kinetics

The concentrations of no degraded vildagliptin remaining after each time of degradation or logarithms of these concentrations were plotted against time of degradation, in order to obtain the equations y = ax + b and the determination coefficients R^2^, and in consequence to determine the reaction order. The slopes of the obtained linear equations were used for calculate further kinetic parameters, i.e., degradation rate constant (k) and degradation time of 50% (t_0.5_).

### 3.3. LC-UV Method for Quantitative Measurements

#### Chromatographic Conditions

The analysis was performed with a model 515 pump, a Rheodyne 20 µL injector and a model 2487 UV DAD detector controlled by Empower 3 software, all from Waters, Elstree, England). The separation was carried out on a Purospher^®^ RP-18 endcapped column (125 × 4.0 mm, 5 µm) from Merck. The mobile phase was a mixture of 2 mM ammonium acetate-acetonitrile (80:20, *v/v*) with the flow rate of 1.2 mL/min. The UV detection was set at 210 nm. Selectivity of the method was examined by the determination of no degraded vildagliptin in the presence of its degradation products as well as by the determination of vildagliptin in the presence of excipients from the powdered tablets.

### 3.4. Analysis of Stressed Samples by LC-UV Method

After acidic, basic and oxidative degradation, the samples were immediately cooled, neutralized if necessary and diluted to 3.0 mL. Then, 1.5 mL volumes were dispensed to 5 mL volumetric flasks, mixed with 0.20 mL volumes of the I.S. stock solution (1 mg/mL), diluted to the mark with methanol and analyzed using the LC-UV method described above. Concentrations of the remaining (no degraded) vildagliptin were calculated from the mean calibration equation while percentage degradation was calculated from the initial concentration of the drug.

### 3.5. UHPLC-DAD-MS Analysis

UHPLC-DAD-MS/MS was performed using a Dionex Ultimate 3000RS device (Dionex, Sunnyvale, CA, USA). The chromatograph was coupled with a Bruker Amazon SL ion trap mass spectrometer (Bruker Daltonik, Bermen, Germany) without splitting. The separation was carried out using a Kinetex XB-C18 column (150 × 2.1 mm, 1.7 µm) from Phenomenex (Torrance, CA, USA). The column was eluted with A: 0.1% formic acid in deionized water and B: 0.1% formic acid in acetonitrile. The two-step gradient was used from 0% B to 30% B in 13 min and 30% B to 65% B up to 20 min. The column temperature was maintained at 25 °C with the flow rate set to 0.3 mL/min. The UV/VIS signal was recorded from 190 to 450 nm. The eluate was introduced directly to the mass spectrometer. The mass spectrometer was equipped with an ESI interface working in a positive ion mode with settings as follows: capillary voltage 4500 V, endplate offset 500 V, nebulizer pressure 40 psi, drying gas temperature 145 °C and gas flow rate 9 L/min. The instrument was used with the smart parameter setting (SPS) fixed at 450 amu. The scan range was from *m*/*z* 70 to *m*/*z* 2200. The MS/MS was switched on and two the most abundant ions were subjected to fragmentation.

### 3.6. Stability of Vildagliptin in the Presence of Excipients

Solid mixtures of vildagliptin and four excipients, i.e., LAC, MAN, MGS and PVP were prepared by mixing the components in an agate mortar at 1:1 ratio (*w/w*). The prepared mixtures were dispersed as ca. 20 mg portions to standardized small flat vessels (the thickness of the layer was approximately 3 mm) and placed in a climate chamber KBF-LQC (Binder GmbH, Tuttlingen, Germany) at 60 °C and 70% RH for 60 days.

Mid-IR spectra were recorded on a Nicolet 6700 spectrometer (Thermo Scientific, Waltham, MA, USA), equipped with a Smart iTR accessory. Reflectance NIR spectra were measured using a Near IR Integrating Sphere from Thermo Scientific. After recording background spectra, the samples weighing ca. 2 mg were analyzed over the range 4000–800 cm^−1^ as mid-IR spectra and over 10,000–4000 cm^−1^ as NIR measurements. Each spectrum was an average of four scans with a resolution of 4 cm^−1^.

All chemometric computations were performed using a free GNU R environment, version 3.4.0. (R Foundation for Statistical Computing, Vienna, Austria). The spectra were standardized with Standard Normal Variate (SNV) algorithm to remove random shifts and random intensity changes, and to preserve only the shape information. PCA was done without scaling because it is a standard strategy during spectral data analysis.

## 4. Conclusions

The results presented here complemented the current knowledge about chemical stability of important antidiabetic drug vildagliptin. Due to the lack of respective data, the results presented here are a valuable supplement to the literature resources. To reach more definite conclusions, a few analytical techniques, i.e., LC-UV, UHPLC-DADS-MS, mid-IR and NIR spectroscopy with chemometric assessement (PCA) were applied. For the first time, kinetic parameters were calculated for degradation of vildagliptin in solutions of different pH and oxidative conditions. In addition, three new degradants of vildagliptin were detected. It could be especially important because vildagliptin and its related compounds had not been mentioned in any of pharmacopoeias so far. Finally, the stability of vildagliptin was examined in the presence of four excipients with different chemical properties and all excipients were showed to be reactive. Thus, the final pharmaceutical formulations containing vildagliptin should be projected and manufactured with special regard to optimal choice of excipients, as well as storage conditions.

## Figures and Tables

**Figure 1 molecules-26-05632-f001:**
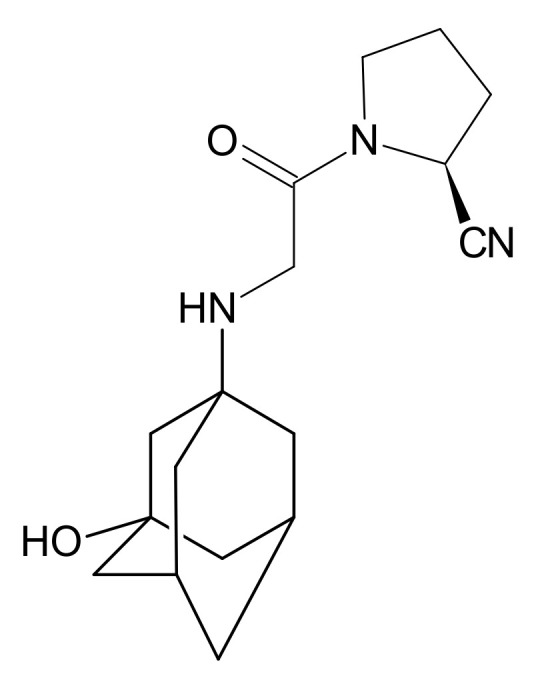
Chemical structure of vildagliptin ((*2S*)-1-{[(3-hydroxytricyclo[3.3.1.13,7]-decan-1-yl)amino]acetyl}pyrrolidine-2-carbonitrile).

**Figure 2 molecules-26-05632-f002:**
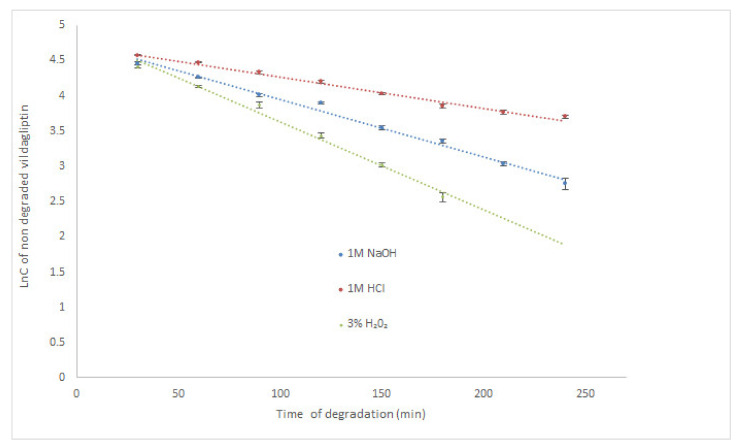
Pseudo first-order plots of degradation of vildagliptin in solutions at 23 °C (mean ± SD, *n* = 3 for each time point).

**Figure 3 molecules-26-05632-f003:**
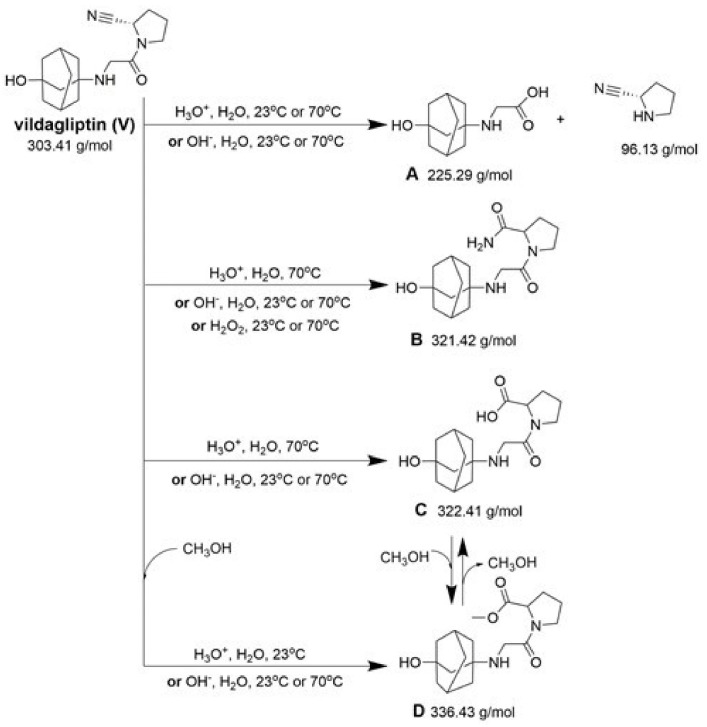
Products of degradation of vildagliptin in acidic, basic and oxidative conditions.

**Figure 4 molecules-26-05632-f004:**
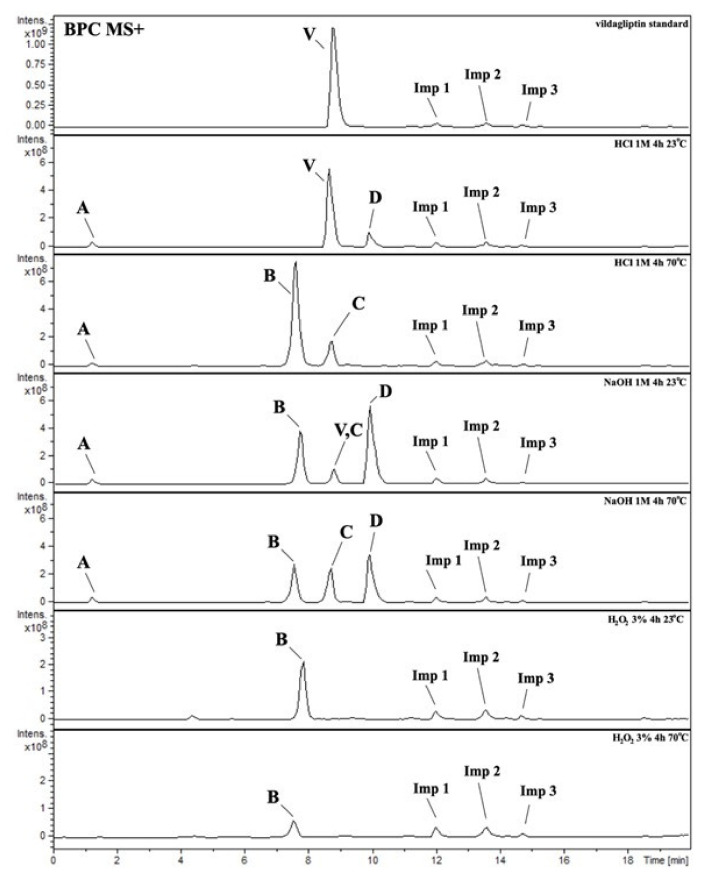
BPC MS+ chromatograms of the analyzed samples. V=vildagliptin, A, B, C, D = Compounds A–D (degradation products of vildagliptin) and Imps 1–3.

**Figure 5 molecules-26-05632-f005:**
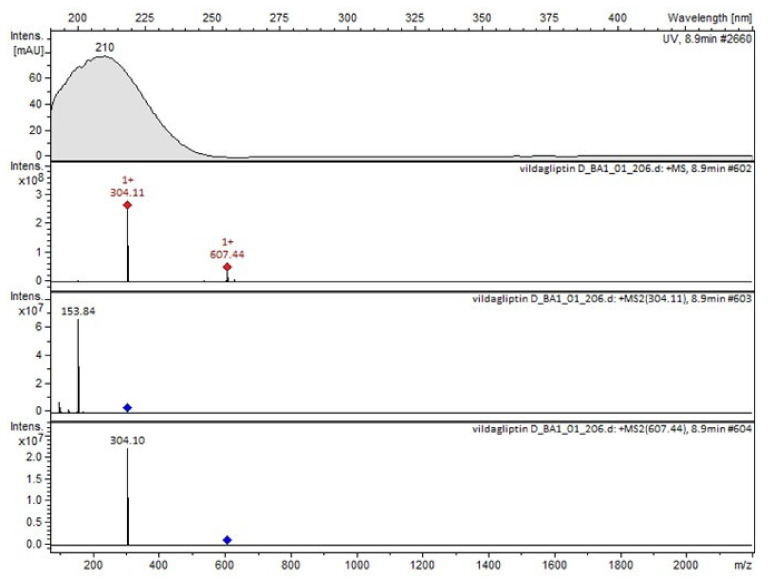
MS and UV/VIS spectra of vildagliptin (V).

**Figure 6 molecules-26-05632-f006:**
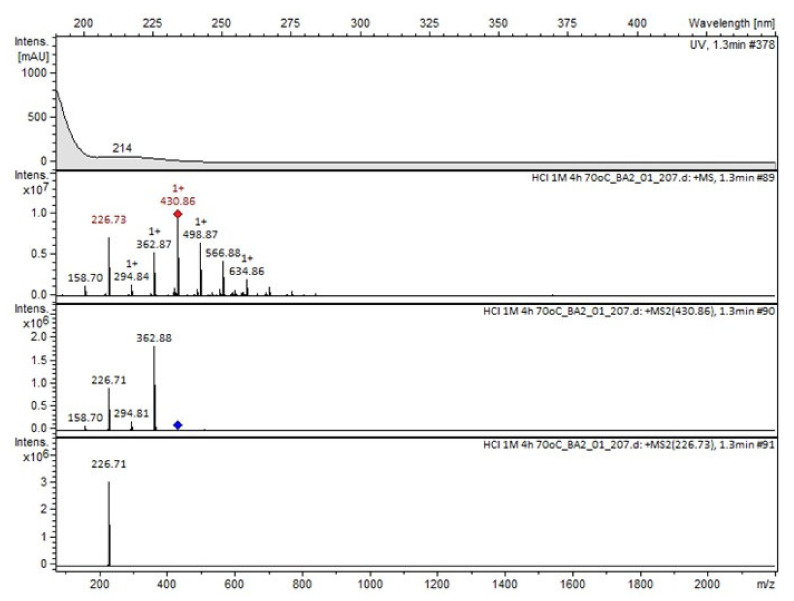
MS and UV/VIS spectra of Compound A.

**Figure 7 molecules-26-05632-f007:**
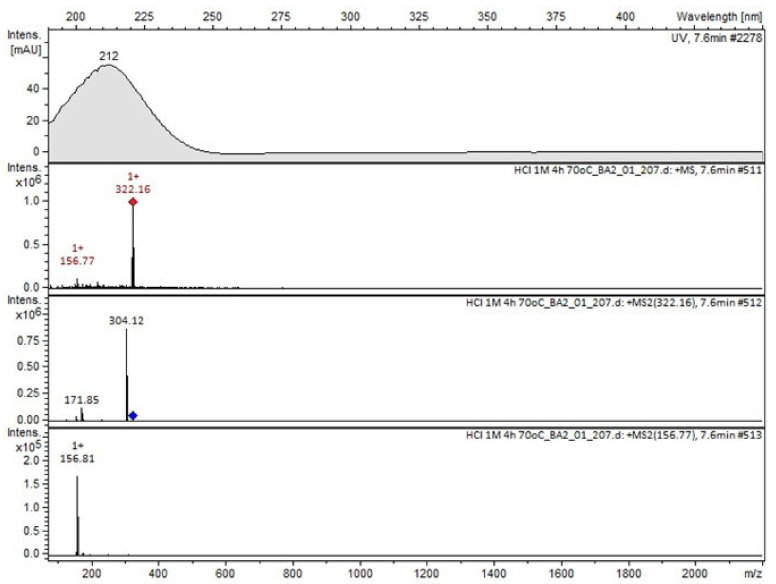
MS and UV/VIS spectra of Compound B.

**Figure 8 molecules-26-05632-f008:**
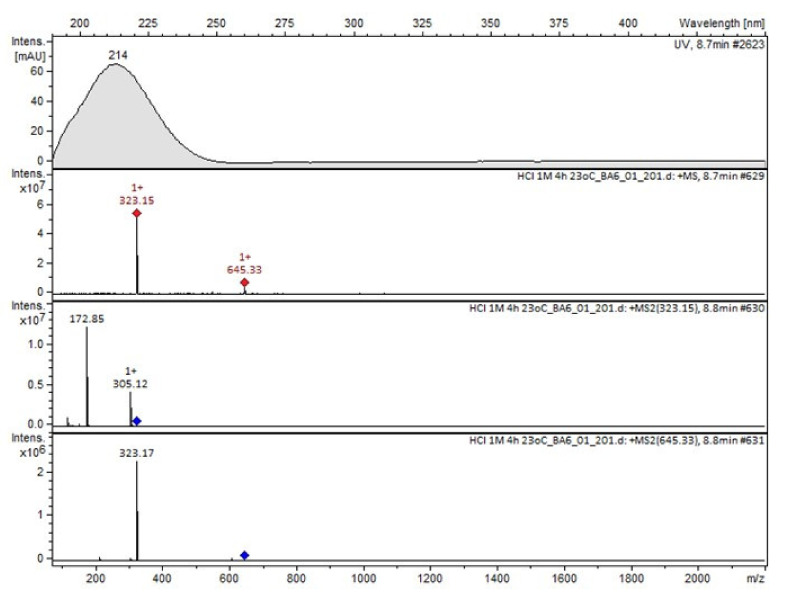
MS and UV/VIS spectra of Compound C.

**Figure 9 molecules-26-05632-f009:**
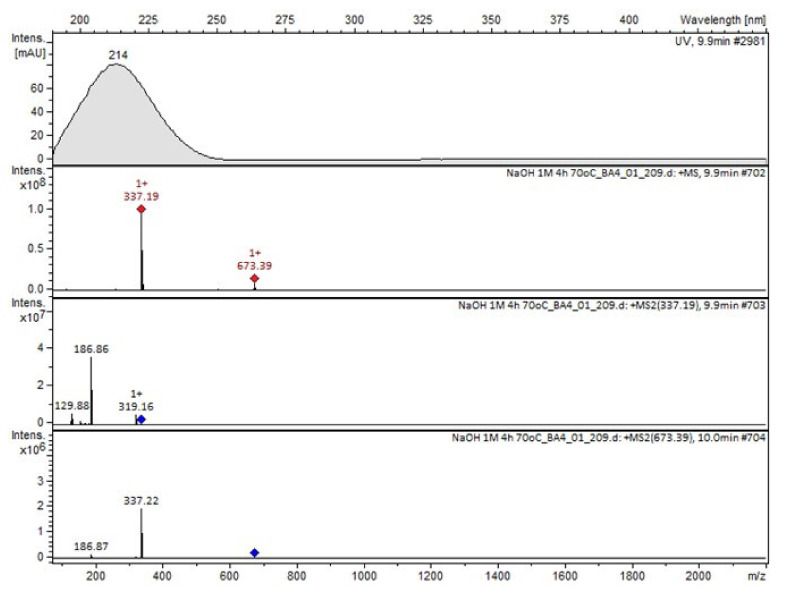
MS and UV/VIS spectra of Compound D.

**Figure 10 molecules-26-05632-f010:**
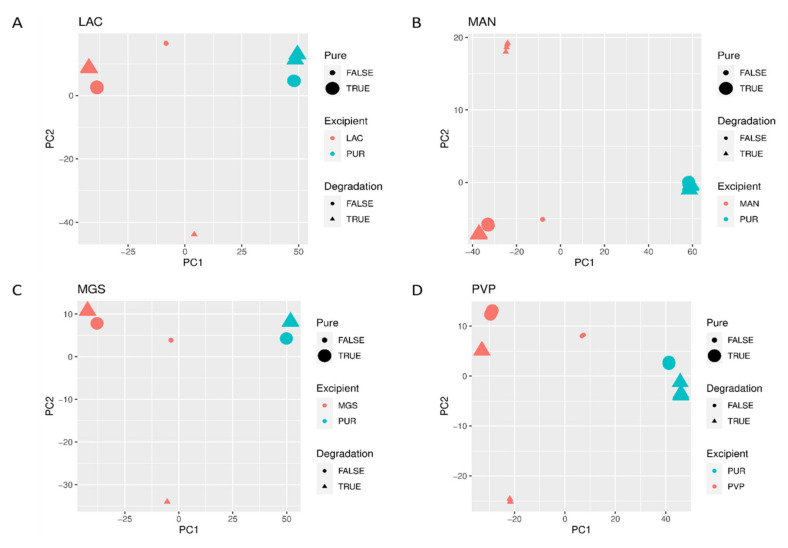
(**A**–**D**) PCA scores plots of mid-IR spectra of vildagliptin (Pure Drug) and the mixtures of vildagliptin with Excipients: (**A**) lactose (LAC), (**B**) mannitol (MAN), (**C**) magnesium stearate (MGS) and (**D**) polyvinylpirrolidone (PVP).

**Figure 11 molecules-26-05632-f011:**
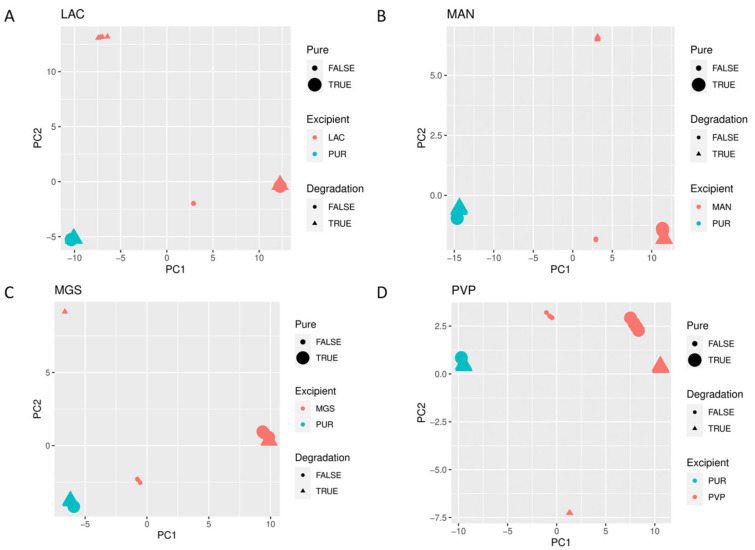
(**A**–**D**) PCA scores plots of NIR spectra of vildagliptin (Pure Drug) and the mixtures of vildagliptin with Excipients: (**A**) lactose (LAC), (**B**) mannitol (MAN), (**C**) magnesium stearate (MGS) and (**D**) polyvinylpirrolidone (PVP).

**Figure 12 molecules-26-05632-f012:**
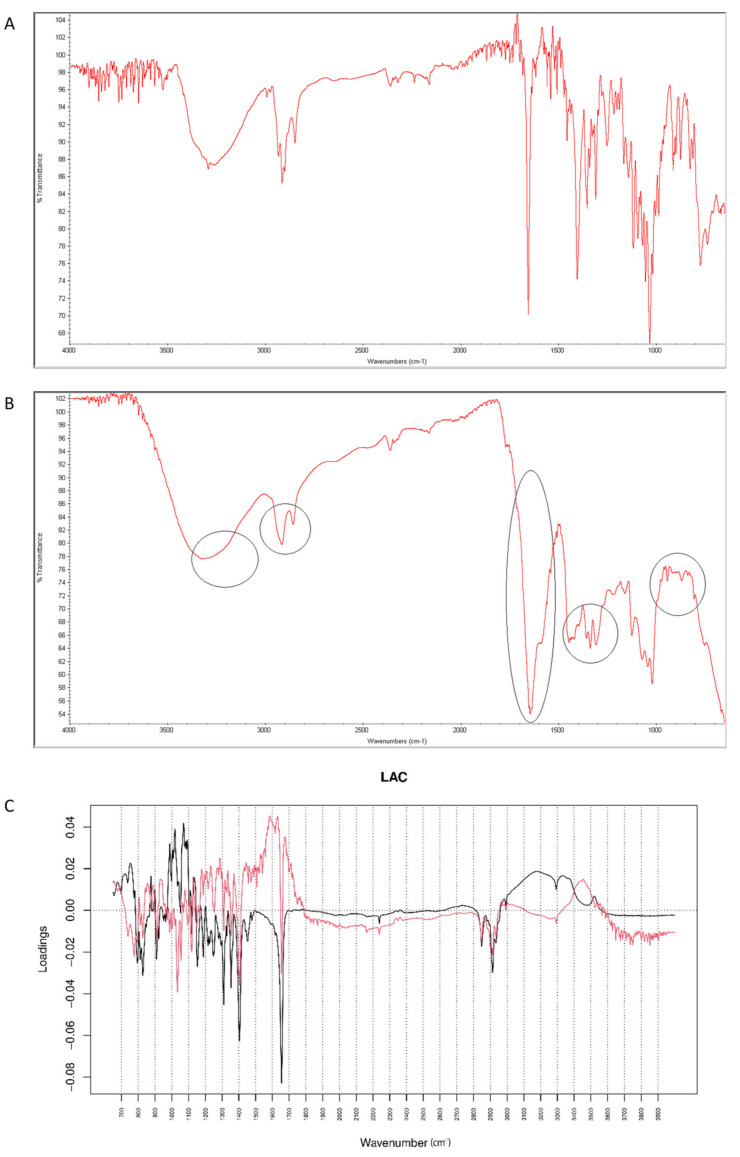
(**A**–**C**) Vildagliptin and lactose (LAC): mid-IR spectra of no stressed mixture (**A**), stressed mixture (**B**) and PCA loadings of the spectra (**C**); PC1 loadings are marked black while PC2 loadings are marked red; most visible differences in the spectrum of the stressed mixture are indicated with circles.

**Figure 13 molecules-26-05632-f013:**
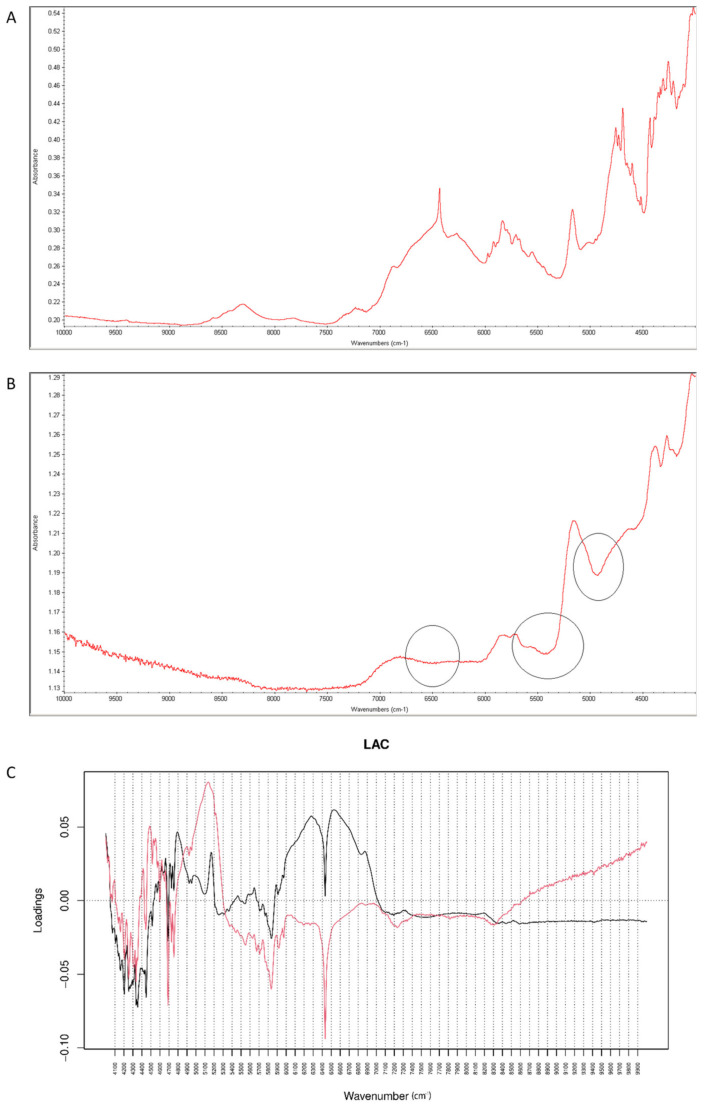
(**A**–**C**) Vildagliptin and lactose (LAC): NIR spectra of no stressed mixture (**A**), stressed mixture (**B**) and PCA loadings of the spectra (**C**); PC1 loadings are marked black while PC2 loadings are marked red; most visible differences in the spectrum of the stressed mixture are indicated with circles.

**Figure 14 molecules-26-05632-f014:**
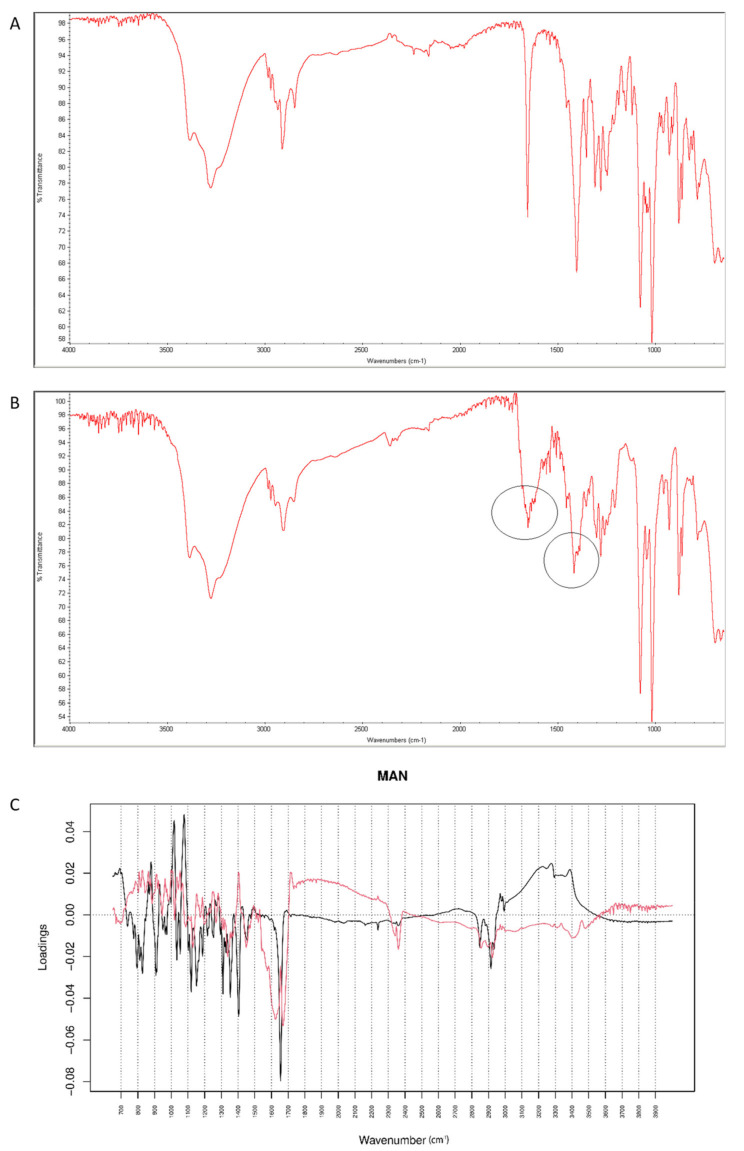
(**A**–**C**) Vildagliptin and mannitol (MAN): mid-IR spectra of no stressed mixture (**A**), stressed mixture (**B**) and PCA loadings of the spectra (**C**); PC1 loadings are marked black while PC2 loadings are marked red; most visible differences in the spectrum of the stressed mixture are indicated with circles.

**Figure 15 molecules-26-05632-f015:**
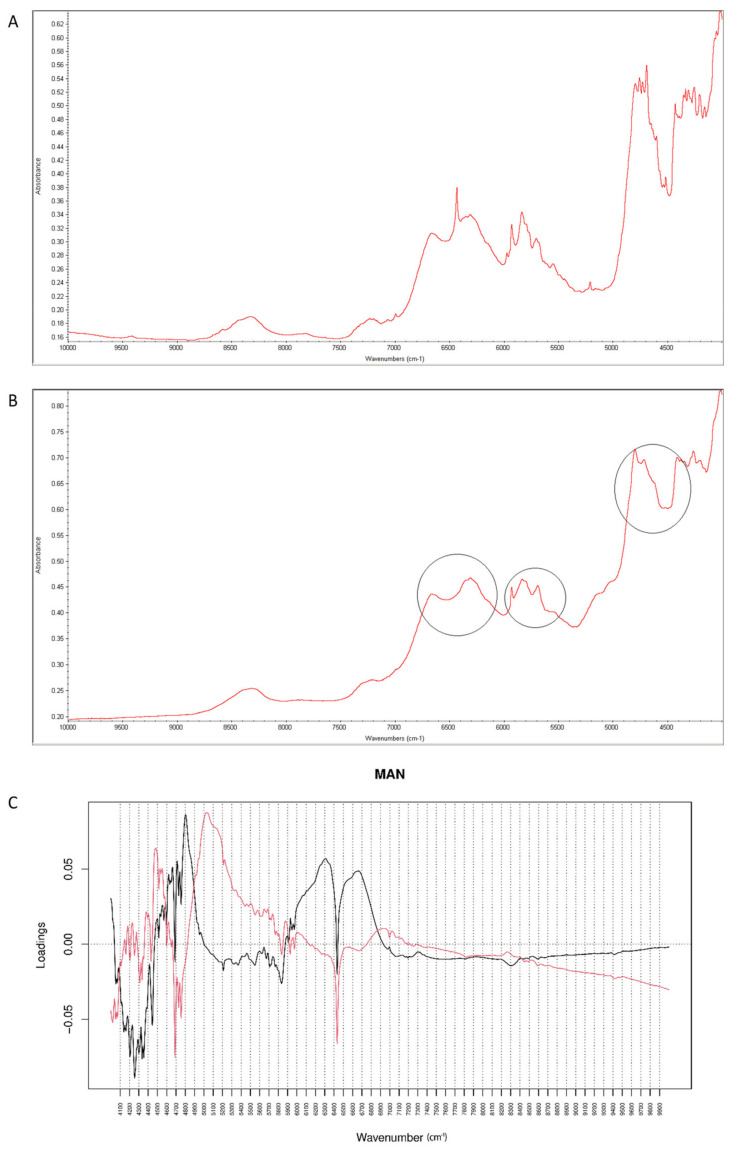
(**A**–**C**) Vildagliptin and mannitol (MAN): NIR spectra of no stressed mixture (**A**), stressed mixture (**B**) and PCA loadings of the spectra (**C**); PC1 loadings are marked black while PC2 loadings are marked red, most visible differences in the spectrum of the stressed mixture are indicated with circles.

**Figure 16 molecules-26-05632-f016:**
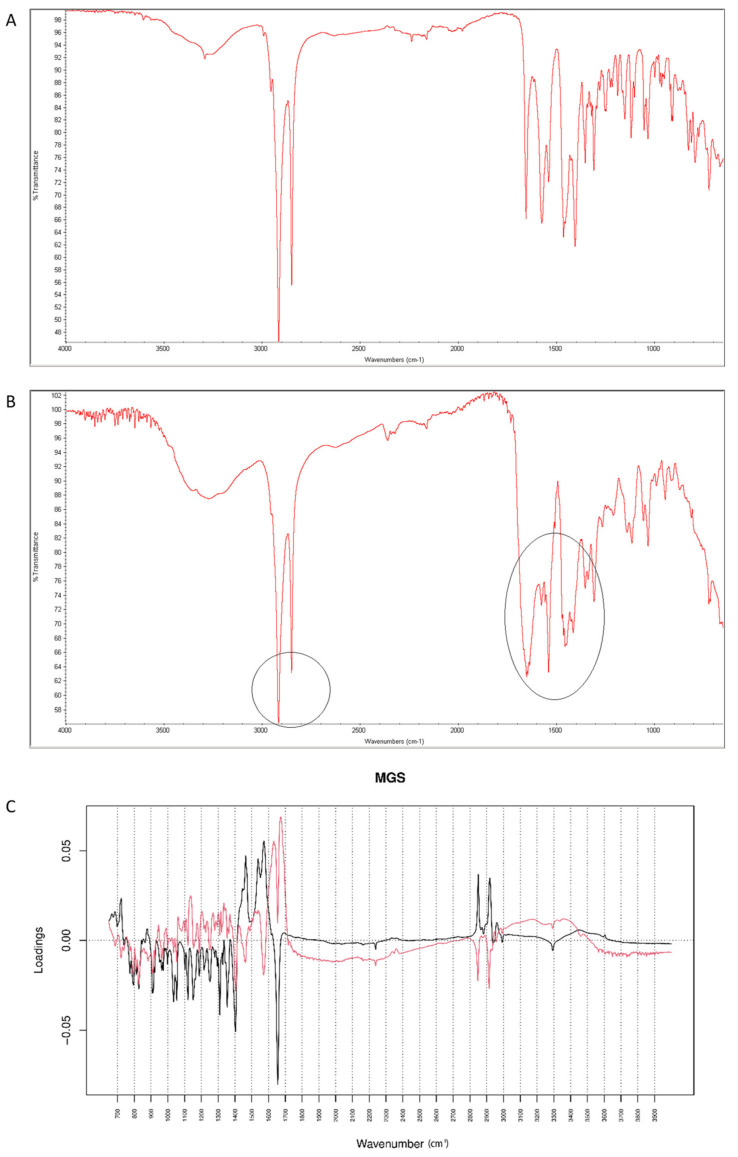
(**A**–**C**) Vildagliptin and magnesium stearate (MGS): mid-IR spectra of no stressed mixture (**A**), stressed mixture (**B**) and PCA loadings of the spectra (**C**); PC1 loadings are marked black while PC2 loadings are marked red, most visible differences in the spectrum of the stressed mixture are indicated with circles.

**Figure 17 molecules-26-05632-f017:**
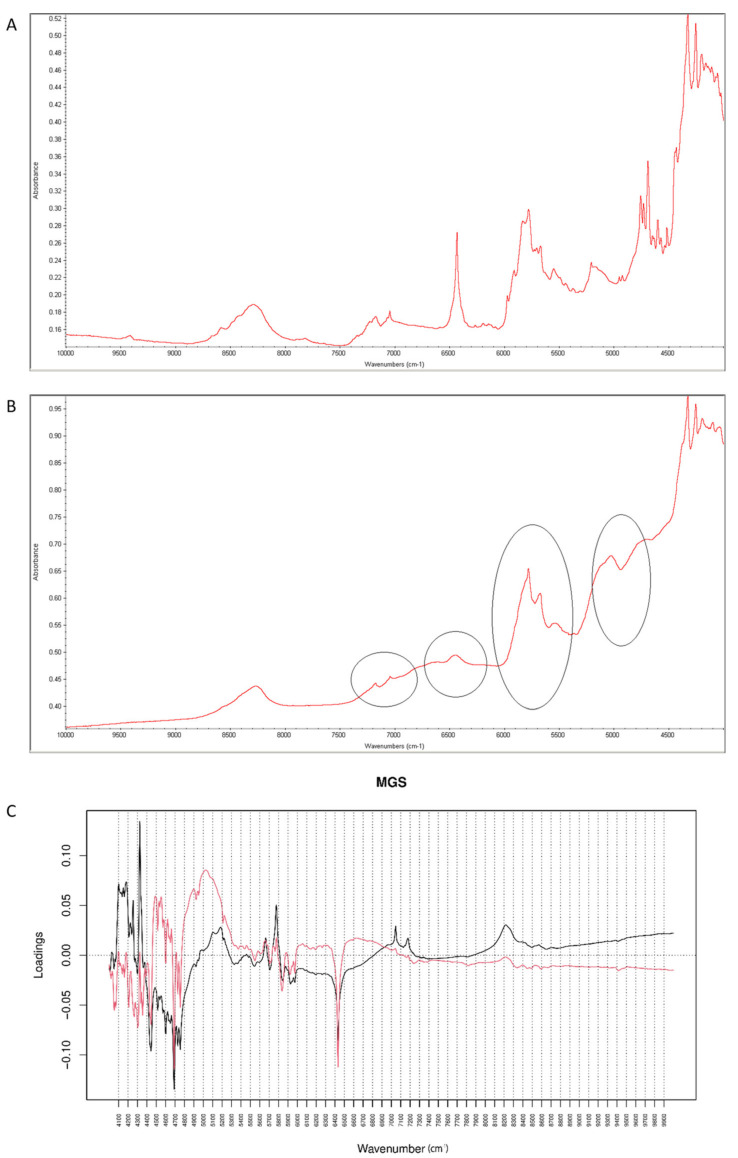
(**A**–**C**) Vildagliptin and magnesium stearate (MGS): NIR spectra of no stressed mixture (**A**), stressed mixture (**B**) and PCA loadings of the spectra (**C**); PC1 loadings are marked black while PC2 loadings are marked red; most visible differences in the spectrum of the stressed mixture are indicated with circles.

**Figure 18 molecules-26-05632-f018:**
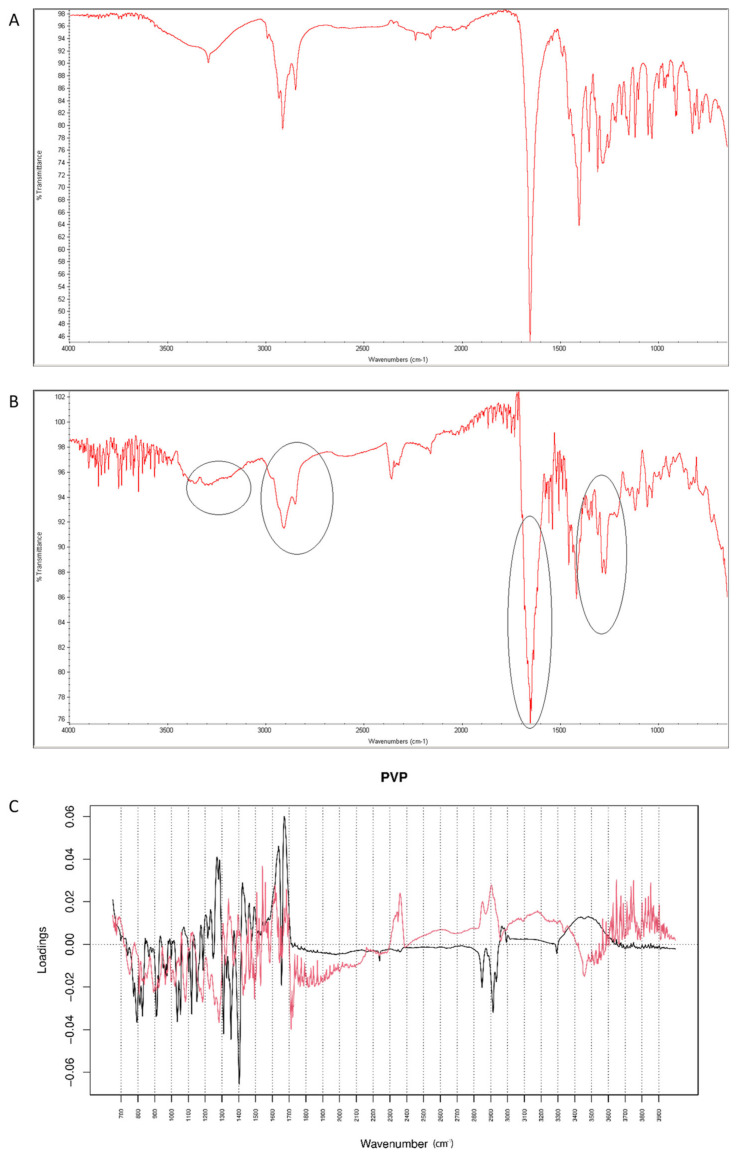
(**A**–**C**) Vildagliptin and polyvinylpirrolidone (PVP): mid-IR spectra of no stressed mixture (**A**), stressed mixture (**B**) and PCA loadings of the spectra (**C**); PC1 loadings are marked black while PC2 loadings are marked red; most visible differences in the spectrum of the stressed mixture are indicated with circles.

**Figure 19 molecules-26-05632-f019:**
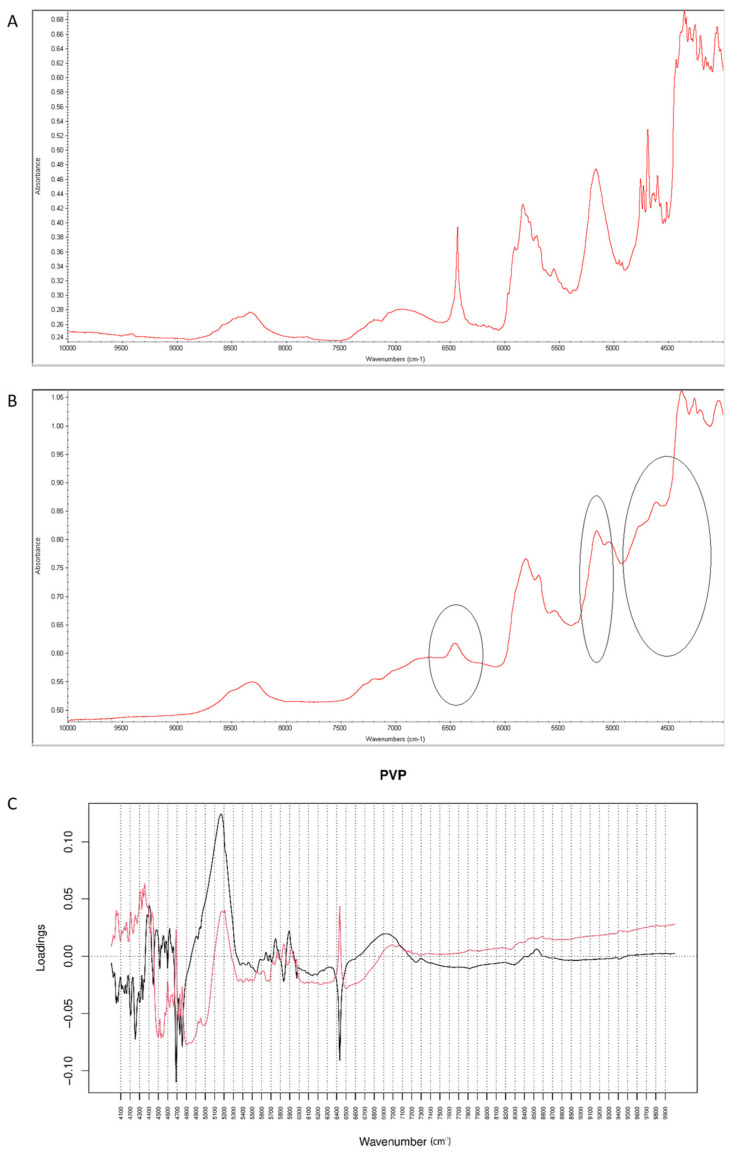
(**A**–**C**) Vildagliptin and polyvinylpirrolidone (PVP): NIR spectra of no stressed mixture (**A**), stressed mixture (**B**) and PCA loadings of the spectra (**C**); PC1 loadings are marked black while PC2 loadings are marked red; most visible differences in the spectrum of the stressed mixture are indicated with circles.

**Table 1 molecules-26-05632-t001:** Parameters of LC-UV method for the quantitative determination of vildagliptin.

Parameter	Results
Linearity range (µg/mL)	40–190
Slope (*n* = 6)	0.00618
SD of the slope	0.000075
Intercept (*n* = 6)	0.02613
SD of the intercept	0.005617
R^2^ (*n* = 6)	0.9997
SD of the R^2^	0.00013
LOD (µg/mL)	2.99
LOQ (µg/mL)	9.09
Accuracy (% Recovery) (*n* = 6)	99.86
SD of the Recovery	1.36
Intra-day precision (% RSD) (*n* = 3)	0.26–0.55
Inter-day precision (% RSD) (*n* = 9)	0.64–1.46

**Table 2 molecules-26-05632-t002:** The percentage level of degradation and kinetic parameters for the degradation of vildagliptin in solutions.

Stress Conditions	Degradation after 240 min [%]	Linear Equation y = ax + b	R^2^	k [s^−1^]	t_0.5_ [h]
23 °C					
1M HCl	59.28	y = −0.0045x + 4.7194	0.9882	1.73 × 10^−4^	1.11
1M NaOH	84.33	y = −0.0081x + 4.7577	0.9907	3.11 × 10^−4^	0.62
3% H_2_O_2_	87.04 *	y = −0.0124x + 4.8789	0.9887	4.49 × 10^−4^	0.40
70 °C					
1M HCl	84.78 **	y = −0.007x + 4.2775	0.9687	2.69 × 10^−4^	0.72
1M NaOH	100	-	-	-	-
3% H_2_O_2_	100	-	-	-	-

* degradation after 180 min; ** degradation after 210 min; k = degradation rate constant; t_0.5_ = degradation time of 50%.

**Table 3 molecules-26-05632-t003:** Related substances of vildagliptin described in the literature.

Compound/Stress Conditions	Structures of Related Substance	[M + H]^+^ *m*/*z*	[Ref.]
IMP A	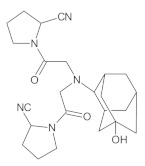	-	[27]
IMP B 1M NaOH, 3% H_2_O_2_	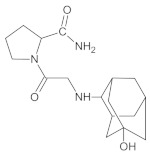	322.1	[20,27]
IMP C	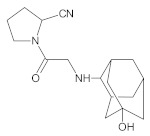	-	[27]
IMP D	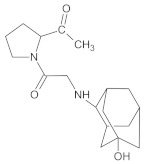	-	[27]
IMP E	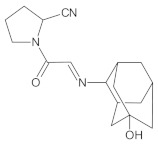	-	[21]
IMP F	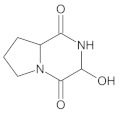	-	[21]
DP 1 0.1M NaOH, 0.3% H_2_O_2_	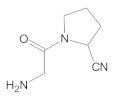	154	[15]
DP 2 1M HCl	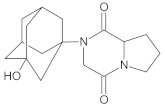	304.0	[20]
DP 3 0.1M NaOH	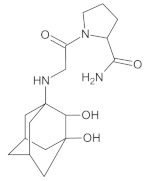	338.2	[20]
DP 4 1M NaOH	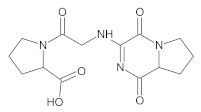	323.6	[20]
DP 5 3% H_2_O_2_	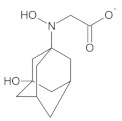	241.0	[20]
DP 6 3% H_2_O_2_	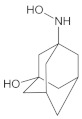	184.3	[20]

**Table 4 molecules-26-05632-t004:** Degradation products of vildagliptin detected by our UHPLC-DAD-MS method.

Degradant/Stress Conditions	Structures and Chemical Names	[M + H]^+^ *m*/*z*
Compound A 1M HCl, 1M NaOH	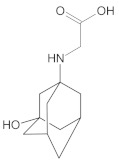	226
	[(3-hydroxytricyclo[3.3.1.1^3,7^]decan-1-yl)amino] acetic acid	
Compound B = IMP B 1M HCl, 1M NaOH, 3% H_2_O_2_	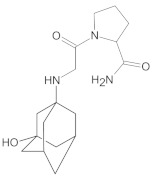	322
	1-{[(3-hydroxytricyclo[3.3.1.1^3,7^]decan-1-yl)amino] acetyl} pyrrolidine-2-carboxamide	
Compound C 1M HCl, 1M NaOH	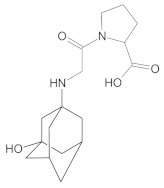	323
	1-{[(3-hydroxytricyclo[3.3.1.1^3,7^]decan-1-yl)amino] acetyl} pyrrolidine-2-carboxylic acid	
Compound D 1M HCl, 1M NaOH	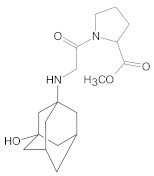	337
	2-*O*-methyl 1-{[(3-hydroxytricyclo[3.3.1.1^3,7^]decan-1-yl)amino] acetyl} pyrrolidine-2-carboxylate	

**Table 5 molecules-26-05632-t005:** UHPLC-DAD-MS2 data of the detected compounds in the analyzed samples.

Compound	t_R_ [min]	UV/VIS [nm]	[M + H]^+^ *m*/*z*	MS^2^ Ions
A	1.3	214	226	159b
B	7.6	212	322	304b, 172, 155
C	8.7	212	323	306, 173b, 116
Vildagliptin	8.9	214	304	154b, 97
D	9.9	214	337	319, 187b, 130
Imp 1	12.0	215	340	322b, 209
Imp 2	13.6	216	453	435b, 322, 209
Imp 3	14.7	216	566	548b, 435, 322, 209

t_R_ = retention time.

**Table 6 molecules-26-05632-t006:** Mid-IR characteristics of vildagliptin, lactose (LAC), mannitol (MAN), magnesium stearate (MGS) and polyvinylpirrolidone (PVP).

Wavenumber [cm^−1^]			Vibrations		
	Vildagliptin	LAC	MAN	MGS	PVP
3100–3500		O-H stretching			
3200–3300			O-H stretching		
3294	O-H stretching				
3294	N-H stretching				
2954				C-H stretching	
2910	C-H stretching			C-H stretching	
2900		C-H stretching	C-H stretching		
2850	C-H stretching				
1658	C=O stretching				
1650					C=O stretching
1577				C=O stretching (COOH)	
1505				C-H stretching	
1495	N-H bending				
1450			C-H bending		
1410					C-H bending
1400					C-C stretching
1300			C-H bending		
1250					C-O stretching
1155	C-O stretching				
1090			C-O stretching		
1040	C-O stretching				
1010			C-O stretching		
1000		C-O stretching			

## Data Availability

Not applicable.

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
