# Peer review of "Comprehensive Insight into Chemical Stability of Important Antidiabetic Drug Vildagliptin Using Chromatography (LC-UV and UHPLC-DAD-MS) and Spectroscopy (Mid-IR and NIR with PCA)"

_molecules, 2021, doi:10.3390/molecules26185632_

Round 1

Reviewer 1 Report

Interesting topic. Useful information for quality control in pharmaceutical companies and especially in the design phase for the choice of suitable excipients.

This work is well presented and easy to read. Experiments were well planned and the analyses were performed by appropriate methods. The results were correctly analysed and interpreted. As a whole, references are pertinent, comprehensive and updated.

It merits publication in Molecules after minor revision. Detailed remarks on the text are as follows:

I would recommend reviewing the abstract and not reporting the used experimental conditions  in detail, for example 70°C, pH 1M HCl and 1M 15
NaOH...

Optimization and validation of quantitative LC-UV method was performed, involving robustness, selectivity, linearity, precision and accuracy. In Materials and Methods it was not specified how robustness, linearity, precision and accuracy were evaluated
